# Prognostic Biomarkers of Systemic Inflammation in Non-Small Cell Lung Cancer: A Narrative Review of Challenges and Opportunities

**DOI:** 10.3390/cancers16081508

**Published:** 2024-04-15

**Authors:** Mark Stares, Leo R. Brown, Dhruv Abhi, Iain Phillips

**Affiliations:** 1Edinburgh Cancer Centre, NHS Lothian, Edinburgh EH4 2XU, UK; 2Cancer Research UK Scotland Centre, University of Edinburgh, Edinburgh EH4 2XR, UK

**Keywords:** non-small cell lung cancer, systemic inflammation, prognosis, biomarker

## Abstract

**Simple Summary:**

Non-small cell lung cancer (NSCLC) is a common diagnosis from which many patients die. Blood tests reflecting systemic inflammation are routinely collected in the NSCLC clinic and may provide information on a patient’s likelihood of a future event. Despite this, these “prognostic biomarkers” are not routinely used in clinical practice. In this narrative review we describe the key biomarkers of systemic inflammation and their prognostic significance in NSCLC. We highlight several challenges that limit their clinical application, including the need to; define an optimal inflammatory biomarker, consider NSCLC as a collection of different diseases, and explore outcomes with respect to how they may change clinical practice. We discuss how these challenges may be overcome through collaboration and the standardisation of recording and reporting of inflammatory biomarker studies. Further, we highlight the potential of modern electronic patient records and advanced data-analyses techniques in this area of research.

**Abstract:**

Non-small cell lung cancer (NSCLC) is a common malignancy and is associated with poor survival outcomes. Biomarkers of systemic inflammation derived from blood tests collected as part of routine clinical care offer prognostic information for patients with NSCLC that may assist clinical decision making. They are an attractive tool, as they are inexpensive, easily measured, and reproducible in a variety of healthcare settings. Despite the wealth of evidence available to support them, these inflammatory biomarkers are not yet routinely used in clinical practice. In this narrative review, the key inflammatory indices reported in the literature and their prognostic significance in NSCLC are described. Key challenges limiting their clinical application are highlighted, including the need to define the optimal biomarker of systemic inflammation, a lack of understanding of the systemic inflammatory landscape of NSCLC as a heterogenous disease, and the lack of clinical relevance in reported outcomes. These challenges may be overcome with standardised recording and reporting of inflammatory biomarkers, clinicopathological factors, and survival outcomes. This will require a collaborative approach, to which this field of research lends itself. This work may be aided by the rise of data-driven research, including the potential to utilise modern electronic patient records and advanced data-analysis techniques.

## 1. Introduction

Lung cancer is the second most common malignancy worldwide, affecting more than two million people each year [1]. It is a devastating disease, with only 40% of affected patients alive at one year after diagnosis [2]. Approximately 80–85% are classified as non-small cell lung cancers (NSCLC) [3]. Clinically, NSCLC may be considered as either localised (i.e., stage I: primary tumour only), locally advanced (i.e., stage II/III: locally invasive or spread to regional lymph nodes), or advanced/metastatic (i.e., stage IV: presence of distant metastatic disease) [4]. As the stage increases more intensive treatments are often indicated. Patients with localised NSCLC usually undergo single-modality treatment with curative intent, using either surgery or radiotherapy, including stereotactic ablative body radiotherapy (SABR) [5]. Multi-modal treatment can improve outcomes for locally advanced disease. In patients with stage III disease, cytotoxic chemotherapy can supplement radiotherapy (i.e., concurrent chemoradiotherapy), and adjuvant therapies, following chemoradiotherapy or surgery, may improve survival outcomes [6]. However, many patients are not suitable for or decline radical therapy [7,8]. For patients with metastatic disease systemic anticancer therapies (SACT) may be used palliatively, with the aim of controlling cancer burden and improving quality of life and survival [9,10]. Despite advances in treatments across all stages of NSCLC, survival outcomes for this disease remain poor, with 5-year survival rates in the United Kingdom (U.K.) for stage I disease of 55%, dropping to only 5% in stage IV disease [2]. As new therapies evolve, however, even with metastatic disease, some patients are now surviving for years rather than months [11]. 

Predicting outcomes for individual patients with NSCLC is a significant clinical challenge. The term biomarker describes a clinical characteristic or molecular or genetic change that is indicative of a disease process. Biomarkers may provide prognostic information. A prognostic biomarker indicates an increased (or decreased) likelihood of a future clinical outcome. They are measured at a defined baseline, and although they may include a specific background treatment, the information they provide is independent of the treatment received. The clinical outcomes examined may include the likelihood of cancer recurrence, treatment toxicity, or survival.

Inflammation and the immune system play key roles in the development, progression, and management of cancers. As early as the 1860′s, Virchow proposed that inflammation and cancer were linked, with cancer arising at sites of inflammation [12]. More recently, “tumour promoting inflammation” and “avoiding immune destruction” have been described as hallmarks of cancer [13]. At the level of the tumour microenvironment (TME), complex interactions between cancer cells and stromal and inflammatory cells are recognised as important regulators of all stages of tumour growth. The mechanisms that cancers co-opt to evade immune destruction have been exploited for therapeutic gain in the development of immunotherapies such as immune checkpoint inhibitors (ICI). 

The inflammatory TME is reflected in the systemic inflammatory response to cancer, which may be measured using routine clinical investigations. Blood tests, anthropometric measurements and radiological imaging may all provide an indication of systemic inflammation and/or its sequalae, such as cancer cachexia [14,15,16,17,18,19]. In recent years, there has been a growing interest in the relationship between these inflammatory biomarkers and clinical outcomes in patients with cancer. It has become clear that high levels of systemic inflammation, as determined by blood tests taken as part of routine clinical care, are associated with poorer outcomes, irrespective of tumour stage and treatment. Despite the overwhelming evidence to support their prognostic value, these biomarkers rarely influence clinical management.

In this narrative review, we summarise the key biomarkers of systemic inflammation derived from blood tests taken as part of routine clinical care and their prognostic relevance to patients with NSCLC. We also explore the opportunities for their application and highlight challenges that have so far limited their use in routine clinical practice. 

## 2. Biomarkers of Systemic Inflammation

Numerous blood tests are routinely collected during the investigation and management of patients with NSCLC, almost all of which may reflect aspects of the systemic inflammatory response (Table 1). A key benefit of these tests is that they are widely used and utilise readily available collection and analytic techniques, with results returned rapidly. They are inexpensive, repeatable, and standardised, with well-defined internationally recognised units and normal reference ranges. As a result, they may be applied in almost all healthcare services around the world at a patient’s point of contact with a healthcare provider. 

Herein, we consider a selection of important blood-based biomarkers of systemic inflammation with family groups based on the type of cell or protein that is measured. 

## 3. Leukocytes

Leukocytes are critical components of the innate and adaptive immune system. Broadly described as “white cells”, they are routinely measured as part of the full blood count (FBC) either as a whole (i.e., the white cell count (WCC)) or as the constitutive cell types (i.e., neutrophil count (NC), lymphocyte count (LC), basophil count (BC), eosinophil count (EC), and monocyte count (MC)). In the circulation, they provide useful information about systemic inflammation in response to trauma, disease, or infection. These cells also exert immune-stimulating or suppressive effects and play a key role in the inflammatory TME. Each cell type is associated with clinical outcomes in NSCLC. For example, a high baseline EC (≥130/μL) confers a more favourable prognosis in patients with NSCLC treated with ICI therapy [20].

Although these leukocyte biomarkers may be used individually, they are more commonly combined into *ratio scores*. As the most abundant inflammatory cells observed in the TME, the clinical significance of the neutrophil/lymphocyte ratio (NLR) has been the most extensively investigated of all leukocyte biomarkers. Both circulating neutrophilia and lymphopenia are independently associated with poorer prognosis in patients with cancer [15,21]. Neutrophilia is a common feature of cancer-associated inflammation, with neutrophils responsible for the production of cytokines and the suppression of cytotoxic T cells, i.e., states that promote tumour progression and metastatic spread [22,23,24]. By contrast, lymphopenia represents a stunted cell-mediated adaptative immune response that facilitates tumour development and unchecked growth [25]. The NLR therefore reflects the balance between these pro- and anti-tumorigenic effects. A high NLR has been consistently associated with poor patient outcomes in NSCLC [26]. In patients with stage I NSCLC treated with a single fraction of SABR, NLR has been shown to be an independent predictor of progression free survival (PFS), overall survival (OS), and distant (i.e., metastatic) relapse [27]. 

## 4. Platelets and Coagulation Factors

In addition to their role in haemostasis, thrombosis, and wound healing, platelets play an important role in the inflammatory response. They release various cytokines and chemokines that drive the migration of inflammatory cells, enhancing inflammation in the TME, driving cancer growth and metastatic potential [28,29,30]. Moreover, cancer cells may activate platelets and utilise their prothrombotic properties to evade immune detection [31]. These same mechanisms underly the increased risk of thrombosis in patients with cancer. It is therefore not surprising to find that elevated circulating platelets are a recognised predictor of poor prognosis in NSCLC. In operable localised and locally advanced NSCLC, patients with normal pre-operative platelet counts demonstrated better 3-year OS and disease-free survival (DFS) [32]. Interestingly, platelet counts were higher in those patients with larger or more invasive tumours. 

Like the leucocyte biomarkers of systemic inflammation, platelets are most commonly used within ratio scores, typically alongside LC in the platelet/lymphocyte ratio (PLR). An elevated PLR was significantly associated with poor survival outcomes in several meta-analyses of patients with NSCLC across a range of clinical settings [33,34,35]. Qiang et al., Wang et al. [36,37] and Zhou et al demonstrated that, in patients with NSCLC receiving ICI, low PLR was associated with better OS and PFS as well as a higher overall response rate (ORR) and disease control rate (DCR) [33].

Other components of the coagulation cascade hold prognostic value in NSCLC. Levels of fibrinogen increase in inflammatory states as part of the acute-phase response. In patients who underwent surgical resection of stage I NSCLC, those with pre-operative fibrinogen ≥ 377 mg/dl had poorer relapse-free survival (RFS) and OS [38]. D-dimer, prothrombin (PT), activated partial thromboplastin time (APPT), thrombin time (TT), and the international normalised ratio (INR) have also been identified as independent prognostic factors in lung cancer [39,40]. Significantly, a recent meta-analysis by Bayleyegn et al. showed that multiple coagulation factors, including PT, D-dimer, fibrinogen, and platelets, were higher in lung cancer patients compared to controls, suggesting they could even provide a clue for early diagnosis or aid risk stratification [40].

## 5. Albumin

Albumin is the most abundant circulating protein in humans, constituting approximately half of serum protein. These proteins serve many functions, including transporting hormones, drugs, and fatty acids as well as buffering pH and maintaining oncotic pressure. Historically, serum albumin levels were widely considered as biomarkers of malnutrition; however, it is now also well established as an acute-phase protein that is down-regulated as part of the inflammatory response [41,42]. As such, low levels of albumin have been shown to predict poor outcomes in a range of malignant and non-malignant diseases [14,43,44]. With a relatively long half-life of approximately 14–20 days, albumin may be considered a biomarker of chronic inflammation [45].

Our own work has established the prognostic significance of hypo-albuminaemia in patients receiving SACT for NSCLC [14]. Pre-treatment albumin < 35 g/L, a standard cut-off in clinical practice, was associated with a significantly higher risk of death in patients treated with either ICIs or targeted therapies. Serum albumin measurements during treatment and at the time of progressive disease also predicted subsequent survival [46]. Early serum albumin decrease of ≥10% from baseline has been identified as a biomarker of poor response to ICI monotherapy in patients with NSCLC, where it may be associated with the metabolic clearance of these agents [47]. These data suggest that albumin may have a longitudinal role in predicting treatment response. 

Globulin, categorised as either alpha-, beta-, or gamma-globulin, also holds prognostic value in NSCLC. These globulins are produced by either the liver or by components of the immune system and are critical in the immune response and chronic inflammation. Globulin has been investigated as part of the albumin/globulin ratio (albumin/total), which predicts long-term survival in early and late-stage NSCLC settings [48,49,50]. 

## 6. Non-Specific Biomarkers of Inflammation (CRP, LDH, and ESR)

C-reactive protein (CRP), lactate dehydrogenase (LDH), and erythrocyte sedimentation rate (ESR) are non-specific biochemical markers of systemic inflammation. CRP is perhaps the most widely used of these in routine clinical practice. It is an acute-phase protein synthesized by the liver in response to inflammatory stimuli. It exerts pro-inflammatory responses that are associated with tumour invasion and plays a role in promoting tumour immune escape [51].

As an individual biomarker, high levels of CRP are a significant predictor of clinical outcomes in a range of settings in NSCLC. For example, in patients with early-stage NSCLC (i.e., stage I–III) treated with curative intent surgery or SABR, a high baseline CRP was associated with poor OS [52]. In patients with advanced NSCLC treated with cytotoxic chemotherapy, elevated CRP was independently associated with poor response and worse survival, even after adjusting for age, gender, smoking status, and NSCLC pathological subtype [53].

LDH is also frequently measured in patients with cancer. LDH may modulate the tumour microenvironment (TME) by increasing the production of lactate and promoting immunosuppression [54,55]. It is a widely recognised prognostic biomarker in patients with melanoma. As early as the KEYNOTE-001 study, which explored the use of first-line pembrolizumab in patients with advanced melanoma, elevated LDH was found to be associated with higher tumour burden and OS [56]. Similar findings have been observed in patients with NSCLC [57]. The correlation between LDH and tumour burden is of particular interest here. Elevated LDH levels relate to tumour necrosis in the presence of hypoxia, a phenomenon associated with rapid tumour growth [58]. 

## 7. Composite Biomarkers

Numerous other scores have been proposed, combining biomarkers of systemic inflammation from the different families noted above (Table 2). Perhaps the most widely recognised of these *composite biomarker scores* is the modified Glasgow Prognostic Score (mGPS), which combines albumin and CRP [44,59,60]. Originally defined as the GPS but modified in subsequent studies by weighting CRP, this score has been validated in tens of thousands of patients with cancer. It predicts survival, response and resistance to treatment, as well as toxicity and quality of life, including in patients with NSCLC [61,62,63]. These two factors have also been combined as the CRP/albumin ratio (CAR). In patients with locally advanced NSCLC treated with concurrent platinum-based doublet chemoradiotherapy (CRT), an elevated CAR is associated with advanced T-stage, poor PS, worse local control rates, and shorter PFS and OS [64]. 

Our group also proposed the Scottish Inflammatory Prognostic Score (SIPS), combining albumin and NC without weighting [15]. SIPS independently predicts PFS and OS in patients with advanced/metastatic NSCLC expressing programmed death-ligand 1 (PD-L1) ≥ 50%. In describing SIPS, we acknowledge the lack of routine collection of CRP in our region and highlight that albumin and NC, but not CRP, are routinely collected in most clinical trials. Albumin forms a central part of other composite biomarker scores, such as the Prognostic Nutritional Index (PNI), Gustave Roussy Immune Score (GRim), Royal Marsden Hospital Prognostic Score (RMH), platelet/albumin ratio (PAR), CRP–albumin–lymphocyte index (CALLY), and NC/albumin ratio (NAR), all of which have demonstrated prognostic significance in patients with NSCLC [69,73,80,81,82]. Scores incorporating an NLR or derived NLR (dNLR, i.e., NC/(WCC-NC)) backbone have also been developed. Amongst these, the most widely described is the Lung Immune Prognostic Score (LIPI), combining dNLR with LDH [66]. 

Other clinicopathological factors have been incorporated with inflammatory biomarkers to enhance their prognostic value. The advanced lung cancer inflammation index (ALI) takes into account the association between inflammation and body habitus by combing pre-treatment albumin, NLR, and body mass index (BMI) [68]. The Holtzman score includes age, sex, and smoking status, each of which are individually associated with outcomes in patients with NSCLC [83]. The Lung Immuno-Oncology Prognostic Scores, developed specifically for patients treated with ICIs, includes pre-treatment steroid use, an independent prognostic factor in this setting [84]. The drawback of including other clinical data is that they may not be readily available in the clinical record, they often rely on patient-self reporting, and may they be subjective in assessment.

## 8. Challenges

Despite the wealth of evidence supporting the prognostic value of biomarkers of systemic inflammation in patients with NSCLC, they are not yet routinely used within clinical practice. There are a multitude of reasons for this, but herein, we highlight three key challenges that we believe must be overcome to achieve confidence in their application.

## 9. Identifying an Optimal Biomarker of Systemic Inflammation

The high volume of studies investigating biomarkers of systemic inflammation in patients with NSCLC is reflected by the breadth of biomarkers described. However, an optimal biomarker has yet to be confidently determined. This is vital when one considers that different biomarkers may predict a range of outcomes in an individual patient. For example, patient A may be classified as having either low, moderate, or high inflammation dependent on the score used, each with very different prognostic implications [15] (Table 3). This discordance undoubtably limits their clinical application.

The overwhelming majority of published studies in this field focus on a narrow range of potential inflammatory biomarkers, often examining only a single biomarker and its constituent factors. For example, Tang et al. conducted a systematic review and meta-analysis of NLR in patients with NSCLC receiving EGFR-targeted therapy [65]. Of the ten individual studies identified, six investigated only NC and LC, four also included MC, and one included platelets. Although this study provides confidence in the prognostic value of NLR in this setting, it tells us nothing about how NLR performs in comparison to other scores. 

Only a handful of studies have compared biomarkers of systemic inflammation across the different families highlighted above. Xie et al. investigated 16 biomarkers of systemic inflammation, comprising CRP, platelets, NC, LC, and albumin, in patients with NSCLC across all stages and treatment modalities [77]. Individual constituent inflammatory biomarkers or novel scores were not evaluated. The IBI was identified as the optimal biomarker of systemic inflammation for assessing prognosis, 90-day mortality, prolonged hospital admission, and cachexia. The study benefitted from the use of an internal validation cohort, but it has not yet been independently verified. This is another key shortcoming in attempts to define the optimal inflammatory biomarker. It estimated that only 5% of prognostic model studies report external validation [79]. Such validation is necessary to determine a model’s reproducibility and understand how it may be generalised across populations that may differ based on factors such as race, socioeconomic status, and broader healthcare provisions.

This challenge is compounded by the considerable inter-study heterogeneity in inflammatory biomarker cut-off values used to signify low versus high inflammation. This issue is most acute with respect to ratio inflammatory biomarker scores. In the aforementioned meta-analysis by Zhou et al., PLR cut-off values in individual studies in patients with NSCLC varied from 144 to 441.8 [33]. The authors suggest that a cut-off of between 170 and 200 appears to be most useful in assessing prognosis [33,85]. Similar heterogeneity is seen between studies investigating NLR and the lymphocyte/monocyte ratio (LMR). This heterogeneity reflects a desire to create as much differentiation as possible between favourable and poor prognosis groups, but this variability inhibits the ability to validate these findings. Subsequently, this limits their application in routine practice, with clinicians being unsure which cut-off value to use. 

Although many non-ratio inflammatory biomarker scores utilise standard clinical reference ranges to overcome the issue, they are not immune to this weakness. For example, the mGPS utilises the clinically applied reference ranges for both albumin (≥/<35g/L) and CRP (≤/>10mg/L) in its calculation [60]. However, variants of this score have also been proposed, including the high-sensitivity mGPS (Hs-mGPS) based on a cut-off of 3 mg/L for CRP and adjusted mGPS (a-mGPS) based on a cut-off of 39 g/L for albumin [86]. In patients with resectable NSCLC, HS-mGPS appears to have greater prognostic value than GPS, mGPS, NLR, PLR, or PNI [71]. Again, these findings impact confidence in a specific biomarker’s clinical utility. 

## 10. Considering NSCLC as a Heterogenous Disease

A significant confounding factor in defining the optimal biomarkers of systemic inflammation in NSCLC is that it is not a single disease entity but rather a heterogeneous collection of multiple clinicopathological subgroups. At a histopathological level, NSCLC is comprised of three major subgroups (i.e., adenocarcinoma, squamous cell carcinoma (SCC), and large cell carcinoma), each with their own distinct biological characteristics. The TNM stage also differentiates patients based on the pattern of their disease, determining treatment options and correlates with outcomes. With respect to advanced/metastatic disease, molecular characteristics are increasingly used to guide treatment decisions and are themselves associated with diverse clinical characteristics and outcomes (Figure 1). [9,10]. For example, 70% of patients with epidermal growth factor receptor (EGFR) mutant NSCLC develop brain metastases compared to only 38% with EGFR wild-type disease [87]. Amongst patients with PD-L1 expression ≥50% treated with pembrolizumab, outcomes are more favourable in those with PD-L1 expression ≥90% compared to those with PD-L1 expression 50–89% [88,89].

Although we understand the prognostic value of systemic inflammation within clinically important subgroups such as those defined by stage, histopathology, and molecular characteristics, we know little about how the level and type of systemic inflammation differs between them. We would suggest there is an unmet need to better define the *systemic inflammatory landscape* of NSCLC both as a whole, and between and within subgroups of NSCLC. Significantly, there is evidence to support the assumption that this systemic inflammatory landscape may differ between different subgroups. 

Some inflammatory biomarkers such as the mGPS have demonstrated prognostic value in cohorts of patients with localised, locally advanced, or advanced/metastatic NSCLC [44,77,90,91]. The overall level of systemic inflammation observed appears to rise with increasing stage. For example, in a large meta-analysis, Huang et al. found that the SII in patients with NSCLC was significantly higher in patients with locally advanced disease compared to those with localised disease [67]. Xie et al. also found that that a high IBI was associated with advanced pathological stage and that it could differentiate the prognosis of patients with NSCLC with the same pathological stage [77]. It is less clear if the type of systemic inflammation observed differs between stage. It is logical to hypothesise that it may, given that biomarkers of systemic inflammation play roles at different points in tumour development, growth, and metastatic dissemination, as we have described.

The selected biomarkers of systemic inflammation may also offer prognostic information for some cancer types but not others. By comparing 16 biomarkers of systemic inflammation, Song et al. found that the ALI had the best performance in predicting survival in a mixed cohort of patients with NSCLC and SCLC [82]. However, the mGPS was superior in patients with NSCLC, and the glucose/lymphocyte ratio performed better in the SCLC cohort. Similarly, in a large systemic review and meta-analysis, Zhou et al. found that the PLR predicted OS and PFS in patients receiving ICIs for NSCLC but not SCLC [33]. Such differences may also exist between the major histopathological subgroups of NSCLC, but we are not aware of any existing studies that have specifically evaluated this. However, the complexity that exists here has been hinted at. Fu et al. found that SII was prognostic in patients with adenocarcinoma but not those with SCC and in those with solid nodules but not ground-glass opacity [92]. Similarly, in a large meta-analysis, Jin et al. observed that amongst patients with NSCLC treated with ICIs, the proportion of patients with SCC included in a study may impact the prognostic value of NLR [93]. 

Targeted therapies are standard-of-care treatments for patients with actionable molecular aberrations of EGFR, anaplastic lymphoma kinase (ALK), or c-ros oncogene 1 (ROS1) (Figure 1) [10]. In patients without a driver aberration, ICIs form the backbone of first-line SACT regimens either alone in those with PD-L1 expression ≥ 50% or combined with cytotoxic chemotherapy in those with lower levels of PD-L1 expression, where the specific regimens are further determined by the histological subtype [9]. The PLR offers prognostic information in independent cohorts of patients with EGFR-positive NSCLC, ALK-positive NSCLC, or NSCLC without a driver aberration [33,35,94,95]. However, how the level and type of systemic inflammation differs between patients with respect to their molecular status is not known. We note that, in our population, patients with driver-driven NSCLC receiving targeted therapies are less likely to have albumin < 35 g/L than those without a driver aberration treated with immunotherapy regimens (36% versus 48%, respectively (*p* = 0.026)) [14]. Although SIPS predicts outcomes in patients with PD-L1 expression ≥ 50%, we do not yet understand how this correlates with the improved survival seen in patients with the highest PD-L1 expression [15].

This challenge is made more difficult by the observation that some biomarkers of systemic inflammation offer prognostic information with respect to some treatment options but not others. The RMH score was developed to evaluate life expectancy when selecting patients with unselected cancer types for phase I clinical trials utilising cytotoxic chemotherapy or targeted therapies [75,76]. In defining the GRim score, Bigot et al. used the RMH score as a template but found that the number of metastatic sites was not independently associated with poor prognosis in phase I trials of ICI, replacing this with NLR >6, which was [75]. Such findings have been seen in NSCLC populations. The LIPI was developed for patients with NSCLC being treated with ICIs [66]. In a large retrospective analysis, it was found to predict disease control rate, PFS, OS, and disease progression at the first radiological examination in patients treated with ICI. However, the study observed no differences in PFS or OS between the LIPI subgroups in a cohort of patients with NSCLC treated with cytotoxic chemotherapy [66]. However, Minami et al. found that LIPI was an independent prognostic factor for cytotoxic chemotherapy in non-driver-driven NSCLC and EGFR-targeted therapy in EGFR-mutant NSCLC [96].

## 11. Clinically Relevant Outcomes

The overwhelming majority of studies investigating biomarkers of systemic inflammation report survival outcomes, most frequently including RFS/PFS and OS. This may be expected given that these are considered the gold-standard primary outcomes in clinical trials. Median survival times are most frequently reported for individual biomarkers of systemic inflammation. Whilst undoubtably important, these observations rarely, in our opinion, provide information that is readily translated into clinical practice. For example, Banna et al. evaluated the role of NLR in patients with NSCLC expressing PD-L1 ≥ 50% treated with first-line pembrolizumab. NLR, using a cut-off of </≥5, was predictive of PFS and 2-year OS [97]. The authors attempted to improve the score using PDL-1 status, concluding that NLR < 5 + PD-L1 ≥ 80% may represent an easy-to-assess tool to identify patients with favourable outcomes, with a 2-year OS of 81%. In the oncology clinic, these findings may support the use of pembrolizumab in these patients. However, 43% of patients in the unfavourable group (i.e., NLR ≥ 5/PD-L1 < 80%) were also alive at 2 years. In discussions about the potential benefits of treatment, these odds are unlikely to deter clinicians or patients from undertaking therapy; hence, these results have limited clinical impact. 

In our own investigation of this patient group, we found that NLR ≥ 5 predicted a median PFS of 3.5 months and a median OS of 7.6 months [15]. However, SIPS 2 predicted a shorter median PFS (2.5 months) and median OS (5.1 months). Significantly, 54% of patients with SIPS 2 had progressive disease prior to the first planned radiological assessment at 3 months, with only one in four alive at 1 year. These odds may be more relevant for discussions with patients when deciding to pursue active treatment. Even then, we feel clinicians are likely to offer treatment to those who are fit, and patients are likely to move forward with this given the chance that they will derive a benefit, especially when the alternative is no active therapy. In those that do, we suggest early palliative care referrals and additional early assessments of therapeutic benefit to identify treatment futility more promptly. We hypothesise that this approach may improve end-of-life care pathways and quality of life for patients with the poorest prognosis. 

We note that the majority of treatment-specific studies investigating inflammatory biomarkers in NSCLC do so in patients with advanced/metastatic NSCLC. However, in those with localised or localised or locally advanced NSCLC, surgery may be employed with curative intent. In this setting, RFS is an important clinically relevant outcome. Wu et al. examined the prognostic ability of several inflammatory biomarkers in patients undergoing surgical resection at a single centre [98]. The study benefited from a large sample size (*n* = 2066), consideration of other clinicopathological factors, and the use of an internal validation cohort. The NLR, with a cut-off of </≥2.3, was superior to the PNI and PLR in predicting both RFS and OS. The authors suggested that inflammatory biomarkers may guide individualized post-operative SACT and surveillance. Although NLR dichotomized RFS in a statistically significant manner, we note that almost one in three patients in the low-risk group (i.e., NLR ≤ 2.3) experienced relapsed within 5 years. Again, we feel that patients and clinicians are unlikely to find these results clinically significant when considering post-operative follow-up strategies.

Another key shortcoming is that almost all studies investigating the prognostic value of biomarkers of systemic inflammation utilise biomarkers taken prior to treatment to predict subsequent survival. However, longitudinal assessment of inflammation may offer additional clinically useful information. Hypo-albuminaemia at 12 weeks in patients with advanced/metastatic NSCLC treated with SACT predicts subsequent survival independently of pre-treatment albumin status [14]. Significantly, patients were more likely to be hypo-albuminaemic at the time of progressive disease, when a patient’s inflammatory status may assist decisions to embark on second-line SACT [46]. Nassar et al. assessed the longitudinal relationship between treatment exposure, CRP concentrations, tumour size, and outcomes in patients with advanced NSCLC treated with chemotherapy [99]. They found that on-treatment CRP was a stronger predictor of a patient’s disease status compared to pre-treatment measurements. Further, the dynamic change of CRP during treatment was a strong predictor of prognosis. More work is required to understand whether longitudinal assessment of inflammatory biomarkers may assist routine clinical follow-up. Of particular interest is how it may supplement radiological imaging assessments. 

In addition to survival, biomarkers of systemic inflammation may also be associated with other clinically relevant endpoints such as quality of life and treatment toxicity. The latter is of particular importance as a part of balanced discussions with patients about the benefits versus risks of treatment. A significant risk of adverse events may dissuade patients or clinicians from pursuing such treatments, especially when an individual’s primary aim is to maintain quality of life. However, in comparison to studies exploring the prognostic value of biomarkers of systemic inflammation in NSCLC, there is a dearth of studies exploring their association with treatment toxicity in the same populations. In a recent example, Lee et al. demonstrated that pulmonary complications of chemoradiotherapy in patients with locally advanced NSCLC were more frequent in patients with albumin < 33 g/L [100].

Some studies have highlighted the need to consider treatment-related toxicity with respect to the prognostic value of inflammatory biomarkers. Indeed, Lee et al. found that patients who experience pulmonary complications of chemoradiotherapy had significantly worse OS than those who did not [100]. The GPS has been associated with platinum-based chemotherapy toxicity in patients with metastatic NSCLC [101]. Significantly, patients with raised inflammation were more likely to terminate treatment due to toxicity or die from treatment-related toxicity. The authors suggested that this may partly explain poor survival outcomes in these patients. Conversely, several studies have found that low levels of systemic inflammation appear to be associated with a higher incidence of immune-related adverse events (irAE)potentially life-threatening toxicities associated with ICI use [102,103,104]. However, our recent work suggests that the association between irAE risk and inflammation is cofounded by the independent prognostic value of biomarkers of systemic inflammation [105]. 

## 12. Future Directions

Although this list of challenges seems dauting, we highlight that a major reason for these failings is a lack of a broad, standardised approach to the investigation of prognostic biomarkers of systemic inflammation in NSCLC. Further, this same shortcoming may be applied to studies investigating their prognostic role in other tumour types. We note that although internationally recognised guidelines for the reporting of biomarker studies exist, such as the Reporting Recommendations for Tumour Marker Prognostic Studies (REMARK) checklist, these do not fully account for the nuances of inflammatory biomarker studies [106]. Although we do not attempt to define specific reporting guidelines here, we make several recommendations as a starting point for further discussion amongst experts in this field. 

## 13. A Minimum Biomarker of Systemic Inflammation Common Dataset

Researchers in this field should be encouraged to investigate a broad panel of inflammatory biomarkers rather than focussing a narrow range of—or individual—biomarkers to facilitate efforts to define the optimal biomarker of systemic inflammation. As we noted, almost all the inflammatory biomarkers described here are collected as part of routine clinical practice, so results are likely available for other biomarkers within many of the cohorts already reported. Indeed, haemoglobin, WCC, LC, NC, MC, BC, EC, and platelets are typically analysed and reported together as part of the full blood count. It is also likely that biomarkers of systemic inflammation belonging to other families described here are measured during the same blood draw. Therefore, it is feasible to record these results at the same timepoints to facilitate comparisons of the prognostic value of multiple inflammatory biomarkers within an individual study.

We propose that a minimum common dataset of biomarkers of systemic inflammation, including those from different families, be routinely collected and analysed by individual studies. A starting point may be to include all test results necessary to calculate the composite biomarkers described here (Table 2). More comprehensively, all routinely collected blood tests results (Table 1) could be included to facilitate exploration of novel inflammatory biomarkers. Where this is not possible, researchers should routinely report the prognostic value of all biomarkers available to them. In particular, studies investigating specific composite biomarkers should fully evaluate their constituent factors. This recommendation would require no additional data collection and could easily be applied to existing datasets. 

## 14. A Minimum Clinicopathological Common Dataset

Efforts should also be focussed on standardising the clinicopathological data reported within individual studies to allow investigation of the systemic inflammatory landscape with respect to distinct subgroups of NSCLC. As a minimum, we recommend that this includes TNM stage, histopathological subtype, molecular characteristics, PS, and treatment modality. Again, much of this data will already be available for existing studies, to which additional analyses could be easily applied.

Within this, we highlight the potential for researchers to collect and compare data from multiple clinicopathological subgroups. For example, it is likely that groups investigating the prognostic significance of inflammatory biomarkers in patients with advanced/metastatic NSCLC expressing PD-L1 ≥ 50% treated with pembrolizumab also treat patients expressing PD-L1 < 50% with chemoimmunotherapy regimens. It is therefore surprising that there are limited data available for this patient group, let alone an understanding of how the systemic inflammatory landscape differs between these patient cohorts. 

## 15. Optimising Survival Endpoints

Although we continue to advocate for use of median survival endpoints, we encourage researchers to explore the prognostic utility of biomarkers of systemic inflammation with respect to landmark outcomes. In particular, we recommend more emphasis be placed on identifying patients with poor prognosis, in whom different management strategies, including decisions not to pursue active treatment, may be more appropriate. For example, in the context of advanced/metastatic disease, “life expectancy of at least 3 months” is a common inclusion criterion in clinical trials investigating medicinal products. Optimising prognostic biomarkers with respect to 3-month OS may therefore be a valuable tool for clinical practice. Conversely, in patients undergoing potentially curative treatments, a landmark of 1-year DFS/RFS or OS may be appropriate when considering more intensive therapies. Alternatively, employing landmarks based on the risk of recurrence may be more meaningful in terms of post-operative decision making in this patient group. For example, a biomarker that predicts a very low risk of recurrence at 5 years may aid discussions to de-escalate adjuvant therapy and surveillance follow-up. Such landmark analyses could be easily applied to existing datasets and methodology without the need for additional data collection. We note that there are internationally recognised standards for reporting these outcome measures.

## 16. Discussion

More comprehensive studies to define the optimal biomarkers of systemic inflammation and the systemic inflammatory landscape of NSCLC are an unmet need. Ideally, these would include evaluation of individual inflammatory biomarkers, the previously described ratio and composite scores, and exploration of novel inflammatory scores. Optimal cut-offs for each biomarker would be identified and then compared to other biomarkers in well-defined clinicopathological cohorts with regards to their prognostic significance. A better understanding delivered by such studies would undoubtably, in our opinion, allay many of the concerns clinicians have in applying these biomarkers in routine clinical practice.

The implementation of a standardised approach to studies investigating the prognostic value of systemic inflammation in NSCLC will require a collaborative effort to define key data variables within common data models. However, this approach brings several key benefits, not least of which is the ability to more readily validate the findings of individual studies. Additionally, as we highlighted, the current literature is predominantly focused on examining previously identified biomarkers of systemic inflammation, with the majority neglecting to report the individual constituent inflammatory biomarkers that constitute the scores evaluated. A comprehensive, standardised list of potential inflammatory biomarkers would facilitate the identification of other promising, novel biomarkers. These may include novel combinations of existing inflammatory biomarkers. We suggest researchers focus on the development of polytomous scores such as mGPS or SIPS. Both of these categorise patients into one of three groups, reflecting low, intermediate, or high levels of systemic inflammation, and may have greater sensitivity to identify patients with the most favourable or poorest prognosis. 

We also note that the ability to comprehensively define the systemic inflammatory landscape of NSCLC with respect to clinicopathologically important subgroups of NSCLC is likely to be limited by the numbers of patients available at individual centres. However, the nature of the routinely collected data required lends itself to collaborative efforts both nationally and internationally. By standardising data collection and reporting, it will be possible to develop large datasets capable of exploring the differences in the type and level of systemic inflammation important within and between these subgroups. This will enhance our understanding of their prognostic significance and may allow us to better understand how this relates to the underlying biology of NSCLC for further translational research. We point to the significant gains made by international open-access datasets of molecular data, such as that held by The Cancer Genome Atlas (TCGA) programme, as an example of the power of such collaborative efforts [107]. This is a resource for which we feel the inflammatory biomarker research community should strive. 

As we noted, many of the studies described here could use existing data or only require minimal additional data collection to start addressing the challenges raised. These efforts may also be aided by the increasing accessibility of routine clinical data held in modern electronic patient records. In southeast Scotland, programmes such as DataLoch have been developed to support such data-driven research [108]. Across the U.K., Cancer Research U.K. has identified data-driven research as a key research strategy in recent years, mirroring the importance being placed on this theme internationally [109]. Further, we note that the data required for such studies are routinely collected and standardised within cancer clinical trials. 

The potential to combine clinicopathological, molecular, and inflammatory biomarkers in NSCLC to improve on their individual prognostic power is untapped. These analyses may be enriched by the application of predictive modelling, potentially with advanced computational analytics, including artificial intelligence methodology. Such techniques are becoming more readily available for data-driven research and are enhanced by the availability of well-curated, standardised data. Although we highlight that the combination of an inflammatory biomarker with PD-L1 status may enhance the ability to detect patients with the most favourable outcomes, there is a considerable scope to improve on this, which may be facilitated by the availability of large-scale comprehensive datasets. An example of the success of such approaches can be seen in breast cancer. PREDICT Breast is a prognostication tool for early breast cancer developed using U.K. cancer registry data and subsequently validated and implemented in international cohorts. It combines multiple clinicopathological factors—but not biomarkers of systemic inflammation—to provide prognostic information with respect to curative treatment options. The Ajutorium tool employs machine learning technology to national cancer registry data in the U.K. and United States to develop a similar prognostic tool to guide adjuvant therapy in the same population [110]. In renal cell carcinoma (RCC), multiple clinicopathological and inflammatory biomarkers have been combined within the International Metastatic RCC Database Consortium risk score, which is now used to stratify first-line SACT treatment decisions [111].

Finally, whilst we focus on the prognostic value of biomarkers of systemic inflammation here, we note their predictive potential, too. Where prognostic biomarkers provide information on the likelihood of a future clinic event, predictive biomarkers may identify individuals who are likely to respond to a particular intervention. For example, the presence of an EGFR driver mutation predicts response to EGFR-targeted therapy and directs treatment strategies. Clinical trials offer an ideal environment to investigate the prognostic versus predictive nature of inflammatory biomarkers. It is therefore disappointing that none of the recent practice-changing, large phase III randomised clinical trials in NSCLC have reported on this or made the necessary data publicly available. Examples of this approach do exist in other cancer types, though. The CheckMate 743 study of first-line nivolumab plus ipilimumab versus chemotherapy in patients with unresectable malignant mesothelioma reported outcomes associated with the LIPI score [112]. Although it found LIPI to be prognostic for OS, it was not predictive for treatment choice. 

Few studies on the biomarkers of systemic inflammation have been able to explore their predictive value, primarily as they lack suitable comparative control groups. In particular, the current evidence base lacks information on patients who have been deemed unsuitable or chose not to undergo treatment. Given the increasing complexity and number of treatments available for patients with NSCLC, a predictive biomarker that selects treatment pathways would be a valuable tool in the clinic. Although PD-L1 status directs first-line treatment options in metastatic NSCLC, it is imperfect. Patients with metastatic NSCLC expressing PD-L1 ≥ 50% may be treated with either ICI monotherapy or chemoimmunotherapy, but there have been no prospective comparisons between these two strategies with respect to other biomarkers [9]. Mahait et al. examined the prognostic and predictive value of various biomarkers of systemic inflammation in patients with metastatic NSCLC treated with either ICI, ICI + chemotherapy, or chemotherapy alone at a single centre [107]. With the possible exception of SIPS, none of the scores evaluated were predictive of response to a particular therapy. Although disappointing, we note that the association between survival and different treatment modalities guided by multiple clinicopathological factors may be more complex than can be described by a single biomarker. Again, the collection of standardised datasets in NSCLC may support future analyses of the predictive value of biomarkers of systemic inflammation in this disease.

## 17. Conclusions

Biomarkers of systemic inflammation derived from blood tests taken as part of routine clinical care have undoubted value in predicting outcomes for patients with NSCLC in multiple settings. They are an attractive, inexpensive, reproducible tool that can be easily used and interpreted in the lung cancer clinic. Despite this, they have yet to find a firm standing in routine clinical practice. We have herein identified several challenges to meeting this potential, including the need to better define the systemic inflammatory landscape of NSCLC with respect to clinicopathological features and treatments, relating this to clinically useful outcomes. We point to the need for collaborative efforts to standardise the collection and reporting of inflammatory biomarker data and welcome feedback from the broader research community in realising these ambitions. Although we have focussed on NSCLC, these same challenges apply to all other malignancies in which biomarkers of systemic inflammation have been investigated. 

## Figures and Tables

**Figure 1 cancers-16-01508-f001:**
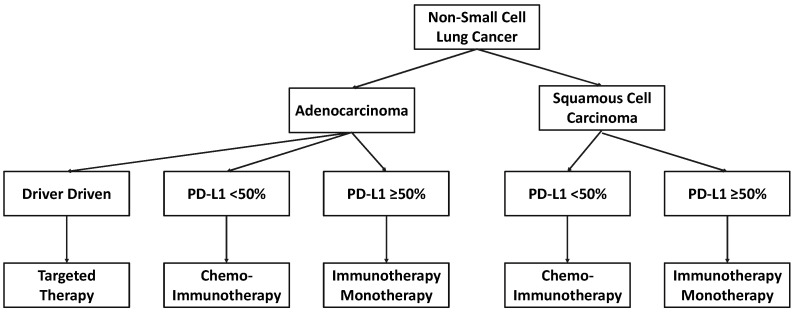
Outline of systemic anticancer therapy algorithm for advanced/metastatic NSCLC.

**Table 1 cancers-16-01508-t001:** Routinely collected blood tests in the oncology clinic. Reference ranges reflect local laboratory thresholds.

Test Item	Reference Range	Units
Haemoglobin	115–165 (female) 130–180 (male)	g/L
Haematocrit	0.40–0.52	Ratio
Red Cell Count	4.5–6.5	×10^12^/L
Mean Cell Volume	78–98	g/L
White Cell Count	4.0–11.0	×10^9^/L
Neutrophil Count	2.0–7.5	×10^9^/L
Lymphocyte Count	1.5–4.5	×10^9^/L
Monocyte Count	0.2–0.8	×10^9^/L
Basophil Count	0.01–0.10	×10^9^/L
Eosinophil Count	0.04–0.40	×10^9^/L
Platelet Count	150–400	×10^9^/L
Activated Partial Thromboplastin Time	21.0–28.0	Seconds
Prothrombin Time	9.0–22.0	Seconds
International Normalised Ratio	0.9–1.2	Ratio
Fibrinogen	1.5–4.0	g/L
Urea	2.5–6.6	mmol/L
Creatinine	64–111	mmol/L
Sodium	135–145	mmol/L
Potassium	3.6–5.0	mmol/L
Phosphate	0.8–1.4	mmol/L
Magnesium	0.7–1.0	mmol/L
Bilirubin	3–21	U/L
Alanine Transaminase	10–50	U/L
Alkaline Phosphatase	40–125	U/L
Calcium	2.20–2.60	mmol/L
Adjusted Calcium	2.20–2.60	mmol/L
Albumin	36–47	g/L
C-Reactive Protein	0–10	mg/L
Lactate Dehydrogenase	125–220	U/L

**Table 2 cancers-16-01508-t002:** Examples of composite biomarkers of systemic inflammation in non-small cell lung cancer described in the literature NC, neutrophil count; LC, lymphocyte count; MC, monocyte count; CRP, C-reactive protein; LDH, lactate dehydrogenase; ULN, upper limit of normal; ECOG PS, Eastern Cooperative Oncology Group Performance Status.

Name	Abbreviation	Calculation
Neutrophil to Lymphocyte Ratio [65]	NLR	NC ÷ LC
Derived Neutrophil to Lymphocyte Ratio [66]	dNLR	NC ÷ (WCC − LC)
Platelet to Lymphocyte Ratio [33]	PLR	Platelets ÷ LC
Monocyte to Lymphocyte Ratio [67]	MLR	MC ÷ LC
Advanced Lung Cancer Inflammation Index [68]	ALI	Body mass index × (albumin ÷ NLR)
Systemic Immune-Inflammation Index [67]	SII	Platelets × NLR
CRP to Albumin Ratio [64]	CAR	CRP ÷ albumin
Platelet to Albumin Ratio [69]	PAR	Platelets ÷ albumin
Neutrophil to Albumin Ratio [70]	NAR	NC ÷ albumin
Albumin to Globulin Ratio [50]	AGR	Albumin ÷ serum globulin
Glasgow Prognostic Score [44]	GPS	1 point each for albumin < 35 g/L, CRP > 10 mg/L; total score 0: low, 1: intermediate, 2: poor
Modified Glasgow Prognostic Score [60]	mGPS	0 = any albumin and CRP ≤ 10 mg/L 1 = albumin ≥ 35 g/L and CRP > 10 mg/L 2 = albumin <35 g/L and CRP > 10 mg/L
High-Sensitivity Glasgow Prognostic Score [71]	HS-mGPS	0 = any albumin and CRP ≤ 3 mg/L 1 = albumin ≥ 35 g/L and CRP > 3 mg/L 2 = albumin < 35 g/L and CRP > 3 mg/L
Adjusted Glasgow Prognostic Score [72]	A-mGPS	0 = any albumin and CRP ≤ 3 mg/L 1 = albumin ≥ 39 g/L and CRP > 3 mg/L 2 = albumin < 39 g/L and CRP > 3 mg/L
Scottish Inflammatory Prognostic Index [15]	SIPS	1 point each for albumin < 35 g/L, NC > 7.5 × 10^9^/L; total score 0: low, 1: intermediate, 2: poor
Prognostic Nutritional Index [73]	PNI	Albumin + (5 × LC)
Systemic Inflammation Response Index [74]	SIRI	NC × MLR
Gustave Roussy Immune Score [75]	GRim	1 point each for: LDH > ULN, albumin < 35 g/L, NLR > 6; score 0–1: low risk; score 2–3: high risk
Royal Marsden Hospital Prognostic Score [76]	RMH	1 point each for: LDH > ULN, albumin < 35 g/L, number of metastatic sites > 2; score 0–1: low risk; score 2–3: high risk
CRP/Albumin/Lymphocyte Ratio [77]	CALLY	(Albumin × LC) ÷ (CRP × 10^4^)
Inflammatory Burden Index [77]	IBI	CRP × NLR
Lung Immune Prognostic Score [66]	LIPI	1 point each for dNLR > 3, LDH > ULN, ECOG PS 1 or 2; total score—0: low, 1: intermediate, 2: poor
Modified Lung Immune Prognostic Score [78]	mLIPI	1 point each for dNLR > 3, LDH > ULN, ECOG PS 1 or 2; total score—0: low, 1: intermediate, 2: poor, 3: very poor
EPSILoN Score [79]	EPSILoN	1 point each for NLR > 4, LDH > 400 mg/dL, liver metastases, smoking < 43 pack-years, ECOG PS ≥ 2; total score—0: low, 1–2: intermediate, 3–5: poor

**Table 3 cancers-16-01508-t003:** Baseline pre-treatment blood tests results and predicted survival outcome in an individual patient with non-small cell lung cancer [15].

	Result	Prognosis	Median Overall Survival
White Cell Count (≤11 × 10^9^/L, >11 × 10^9^/L)	11.0	Favourable	16.8 months
Neutrophil Count (≤7.5 × 10^9^/L, >7.5 × 10^9^/L)	7.67	Poor	6.8 months
Neutrophil/Lymphocyte Ratio (≤5, >5)	3.18	Favourable	20.5 months
Platelet/Lymphocyte Ratio (≤180, >180)	194	Poor	9.9 months
Prognostic Nutritional Index (≥45, <45)	41	Favourable	28.7 months
Albumin (<35 g/L, ≥35 g/L)	29	Poor	7.7 months
Scottish Inflammatory Prognostic Score (0, 1, 2)	2	Very Poor	5.1 months

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
