# Peer review of "Prognostic Biomarkers of Systemic Inflammation in Non-Small Cell Lung Cancer: A Narrative Review of Challenges and Opportunities"

_cancers, 2024, doi:10.3390/cancers16081508_

Round 1

Reviewer 1 Report

Comments and Suggestions for Authors

Stares et al. address the need of prognostic biomarkers of systemic inflammation in NSCLC. However, the importance of prognostic biomarkers in the management of NSCLC is significant. 

There are suggestions that can this paper more impactful.

1. The authors should begin with metabolic heterogeneity and its relevance in prognosis.

2. The authors should highlight intracellular and extracellular prognostic biomarkers in the contexts of NSCLC.

3. The authors should touch upon the conventional and non-conventional biological fluids/materials as sources for prognostic biomarkers.

4. The impact of therapeutic modalities such as surgery should discussed in the context of prognostic biomarkers.

5. Reviewer does not understand the intent for this Sentence "

management of cancers 2024-03-04 16:28:00

6. The updated references are recommended for better timeliness. 

7. The authors present their main intent of the paper with the help of flow models on various forms of biomarkers. 

Comments on the Quality of English Language

Moderate editing. 

Author Response

Reviewer 1

The authors thank reviewer 1 for their time in reviewing this manuscript. A summary of our responses is provided below, with details of changes made to the manuscript highlighted. We welcome further constructive critical appraisal of the manuscript and would be happy to address further comments based on our responses.

Stares et al. address the need of prognostic biomarkers of systemic inflammation in NSCLC. However, the importance of prognostic biomarkers in the management of NSCLC is significant. 

There are suggestions that can this paper more impactful.

  1. The authors should begin with metabolic heterogeneity and its relevance in prognosis.
  2. The authors should highlight intracellular and extracellular prognostic biomarkers in the contexts of NSCLC.
  3. The authors should touch upon the conventional and non-conventional biological fluids/materials as sources for prognostic biomarkers.
  4. The authors present their main intent of the paper with the help of flow models on various forms of biomarkers. 

Points taken. Although the authors recognise that these are additional important prognostic factors in NSCLC, we highlight that the review focusses on biomarkers of systemic inflammation derived from blood tests taken as part of routine clinical care. We have made amendments to the manuscript to make this clearer, including in the abstract and introduction. The authors feel that a description of these additional prognostic biomarkers would not be appropriately comprehensive within the constraints of the manuscript word-count or fully in keeping with the narrative context of this review. However, we welcome comments on any specific points that the reviewer feels should be highlighted.

  1. The impact of therapeutic modalities such as surgery should discussed in the context of prognostic biomarkers.

Many thanks. Further discussion on this theme has been added to the manuscript  - Clinically Relevant Outcomes paragraph 3 and Optimising Survival Endpoints paragraph 1.

5. Reviewer does not understand the intent for this Sentence " management of cancers 2024-03-04 16:28:00

Point taken: This has been highlighted by other reviewers, but this text does not appear in either the original submission or the edited manuscript returned by the editors.

  1. The updated references are recommended for better timeliness. 

Many thanks. Efforts have been made to include contemporaneous references where appropriate and possible. References to guidelines reflect the most recent editions. With regards to specific findings from inflammatory biomarker studies we have tried to include recent relevant studies where possible, or referenced the original paper in which the score was described. As a narrative review we accept that the manuscript does not contain an exhaustive review of the current literature but would welcome feedback on any important work that the reviewer feels is relevant.

Reviewer 2 Report

Comments and Suggestions for Authors

The manuscript focuses on Prognostic Biomarkers of Systemic Inflammation in Non-Small 2 Cell Lung Cancer: Challenges and Opportunities represents a technically correct and timely relevant manuscript available for the publication on this journal after minor suggestions

- In the manuscript, please, could the authors add a brief section integrating inflammation and cancer? How these biomarkers may be combined?

- Could the authors also overview technical approaches eligible to identify other promising biomarkers?

- Please, could the authors also distinguish among predictive, prognostic or recurrent biomarkers? How this definition may differently impact on the role of molecular biomakrers?

- Have the authors evaluated the combination of more than one biomarkers? Please, could the authors add a brief analysis on this theme

Comments on the Quality of English Language

Minor english editing

Author Response

Reviewer 2

The authors thank reviewer 2 for their comments. These have been taken into account and the manuscript amended. A summary of the main changes made is provided below for each point. We feel that these changes have positively impacted on the impact of the manuscript. We look forward to any further feedback.

The manuscript focuses on Prognostic Biomarkers of Systemic Inflammation in Non-Small 2 Cell Lung Cancer: Challenges and Opportunities represents a technically correct and timely relevant manuscript available for the publication on this journal after minor suggestions

- In the manuscript, please, could the authors add a brief section integrating inflammation and cancer? How these biomarkers may be combined?

Point taken. The Introduction has been amended – paragraph 2

- Could the authors also overview technical approaches eligible to identify other promising biomarkers?

Point taken. Amendments have been made to the Discussion – paragraph 2

- Please, could the authors also distinguish among predictive, prognostic or recurrent biomarkers? How this definition may differently impact on the role of molecular biomakrers?

Point taken. This has been considered in the Discussion and amendments made – paragraphs 6 & 7

- Have the authors evaluated the combination of more than one biomarkers? Please, could the authors add a brief analysis on this theme

Point taken. The authors are yet to publish an evaluation of combinations of biomarkers. A review of this theme has been added to the Discussion – paragraphs 2 & 5

Reviewer 3 Report

Comments and Suggestions for Authors

This paper reviews the relationship between inflammation and the prognosis of lung cancer. I don't see many explanations from this point of view, and I think it's a meaningful review. Here are some of the criticisms:

1.   The review article should show the inclusion and exclusion criteria of the citation and provide a flow chart for the selection of citations. Therefore, since this review is narrative, I think it would be better to change the title to Prognostic Biomarkers of Systemic Inflammation in Non-Small Cell Lung Cancer: A Narrative Review for Challenges and Opportunities.

2.   Line 50 "2024-03-04 16:28:00." gibberish. Make sure you understand what it means.

3.   Table 1; Please indicate what Abbreviation means in the legend.

4.   Table 2; Please indicate the number of the citation in the table. 

5.   Table 3; Please indicate the number of the citation in the table. In addition, please indicate what abbreviation means in the legend for all such as “PS, performance status”.

Author Response

Reviewer 3

The authors thank reviewer 3 for their time and the positive feedback provided. The changes suggested have been made in the manuscript. We hope that this will satisfy the reviewer and welcome any further feedback.

This paper reviews the relationship between inflammation and the prognosis of lung cancer. I don't see many explanations from this point of view, and I think it's a meaningful review. Here are some of the criticisms:

1.   The review article should show the inclusion and exclusion criteria of the citation and provide a flow chart for the selection of citations. Therefore, since this review is narrative, I think it would be better to change the title to Prognostic Biomarkers of Systemic Inflammation in Non-Small Cell Lung Cancer: A Narrative Review for Challenges and Opportunities.

Point taken. the title has been changed to reflect this suggestion and the narrative basis of this review has been made clearer within the manuscript.

2.   Line 50 "2024-03-04 16:28:00." gibberish. Make sure you understand what it means.

Many thanks. This has been highlighted by other reviewers, but this text does not appear in either the original submission or the edited manuscript returned by the editors.

3.   Table 1; Please indicate what Abbreviation means in the legend.

Point taken. The abbreviations have been removed from table 1.

4.   Table 2; Please indicate the number of the citation in the table. 

Point taken. The citation has been added to the legend

5.   Table 3; Please indicate the number of the citation in the table. In addition, please indicate what abbreviation means in the legend for all such as “PS, performance status”.

Point taken. A representative citation has been added for each inflammatory biomarker described. Abbreviations have been noted within the legend.